# Comparison of Commercial Fish Proteins’ Chemical and Sensory Properties for Human Consumption

**DOI:** 10.3390/foods12050966

**Published:** 2023-02-24

**Authors:** Moona Partanen, Kaisu Honkapää, Jaakko Hiidenhovi, Tanja Kakko, Sari Mäkinen, Sanni Kivinen, Ella Aitta, Kati Väkeväinen, Heikki Aisala

**Affiliations:** 1VTT Technical Research Centre of Finland Ltd., Tietotie 2, 02150 Espoo, Finland; 2Institute of Public Health and Clinical Nutrition, University of Eastern Finland, 70210 Kuopio, Finland; 3Food and Bioproducts, Production Systems, Natural Resources Institute Finland (Luke), Myllytie 1, 31600 Jokioinen, Finland; 4Food Sciences, Department of Life Technologies, University of Turku, 20014 Turku, Finland

**Keywords:** fish by-products, lipid oxidation, functional properties, generic descriptive analysis, volatile compounds, nutritional value, chemical characterization

## Abstract

To stop overfishing and meet the protein needs of a growing population, more information is needed on how to use marine by-catches, by-products, and undervalued fish species for human consumption. Turning them into protein powder is a sustainable and marketable way to add value. However, more knowledge of the chemical and sensory properties of commercial fish proteins is needed to identify the challenges in developing fish derivatives. This study aimed to characterize the sensory and chemical properties of commercial fish proteins to compare their suitability for human consumption. Proximate composition, protein, polypeptide and lipid profiles, lipid oxidation, and functional properties were analyzed. The sensory profile was compiled using generic descriptive analysis, and odor-active compounds were identified with gas-chromatography–mass spectrometry-olfactometry (GC-MS/O). Results indicated significant differences in chemical and sensory properties between processing methods but not between fish species. However, the raw material had some influence in the proteins’ proximate composition. Bitterness and fishiness were the main perceived off-flavors. All samples, apart from hydrolyzed collagen, had intense flavor and odor. Differences in odor-active compounds supported the sensory evaluation results. The chemical properties revealed that the lipid oxidation, peptide profile, and raw material degradation are likely affecting the sensory properties of commercial fish proteins. Limiting lipid oxidation during processing is crucial for the development of mild-tasting and -smelling products for human consumption.

## 1. Introduction

Fish is known to be one of the most nutritious foods on the planet [1], and it plays a major part in the future as a protein source as the population and the prevalence of malnutrition increase [2]. However, overfishing of popular species is an increasing problem threatening aquatic resources [3], which urges the fish industry to utilize fishes and their by-products more efficiently. Over 20 million tons of fish are used as feed annually, even though 90% of it is food-grade fish [4]. The majority of fishes used for feed are small fish that are hard to fillet and often have an unpleasant taste, color, or texture [5]. To better utilize small food-grade fishes for human consumption, they can be processed into fish protein with, e.g., enzymatic hydrolysis, heat treatment, or pH-shift isolation [6]. In addition to small fish, fish by-products can be used as raw material for fish proteins. By-products can cover up to 70% of fish and they are often used as feed [7].

Despite fish proteins being excellent protein sources for humans, the demand is still low. This is likely due to their fishy odor and taste [6]. Although several studies have been conducted about fish proteins, the information is dispersed, and the production methods and raw materials vary between studies. The study by Nisov et al. [8] compared the chemical, functional, and sensory properties of fish proteins that were enzymatically hydrolyzed and pH-shift isolated from two different fish species. However, the study did not investigate explanatory factors for sensory challenges, such as volatile compounds or lipid oxidation. Similarly, the study by Aspevik et al. [9] compared hydrolysates prepared from the heads and backbones of three fish species for their suitability in food products but did not consider functional properties and only examined peptide size as an explanatory factor for sensory evaluation results. To the best of our knowledge, there is currently no study that has analyzed the sensory profile of fish proteins using GC-MS-O in combination with sensory analysis. Additionally, most studies to date have only analyzed fish proteins produced at a laboratory scale. Therefore, a study on commercial fish proteins is necessary for the further development of food-grade fish proteins.

This study aimed to characterize the chemical, functional, and sensory properties of commercial fish proteins and protein derivatives (fish concentrates, fish hydrolysates, and fish collagen) to compare the suitability of different production methods and fish species for human consumption as well as evaluate the processes causing sensory properties that are limiting the demand of food-grade fish proteins. Explaining factors such as free amino acids, lipid oxidation, and volatile compounds were analyzed to better understand the usability of these proteins.

## 2. Materials and Methods

### 2.1. Materials

In April 2022, efforts were made to procure a large sample set of food-grade fish proteins from multiple manufacturers. However, the availability of such proteins remains constrained in the industry. A sample set of six commercial fish proteins could be acquired from reputable European manufacturers of fish proteins, except the hydrolyzed fish collagen was purchased directly from the retail store. The processing methods and fish species and parts used for the protein are presented in Table 1. Altogether, three fish concentrates (CONC1–3), two hydrolysates (HYDR1–2), and one hydrolyzed fish collagen (COLL1) were analyzed. All the samples were dried protein powders. Samples were received through the postal service at room temperature. After arrival at the research facilities, the samples were stored at −18 °C. The pH of the samples was measured before analyses after 30 min of mixing in a 5% water solution. To demonstrate the behavior of fish protein in use, the pH of all samples was left neutral (ranging from 6.4 to 7.1) without any adjustment.

### 2.2. Proximate Composition

Crude protein was determined with the Dumas combustion method using rapid MAX N exceed equipment (Elementar Analysensysteme GmbH, Langenselbold, Germany), where a conversion factor of 5.58 was used for nitrogen as recommended by Mariotti et al. [10] for the amino acid distribution of fish. For hydrolyzed collagen, 5.4 was used as a conversion factor following the earlier report of Ardekani et al. [11] for the protein content of catfish derived gelatin. Analyses were performed in triplicate.

Lipids of all samples except for COLL1 were extracted as reported by Damerau et al. [12] with slight modifications. Briefly, 3–6 g of fish protein powder was suspended with 7 mL of 8.8% potassium chloride and the mixture was homogenized with an Ultra-Turrax (T 25 digital ULTRA-TURRAX^®^, IKA^®^-Werke GmbH & Co. KG, Staufen, Germany) for 2 min. After the addition of 12 mL of 2-propanol and 12 mL of hexane, the mixture was homogenized again for 2 min, after which 12 mL of hexane was added. The mixture was vortexed for 1 min, and centrifuged (4 °C, 3000 rpm, 20 min), after which the upper phase was collected. The extraction was repeated with 30 mL hexane, and the organic phases of both extractions were collected. The collagen sample, due to its extremely low lipid content, was extracted using a Bligh and Dyer [13] method, as reported by Ozogul et al. [14]. The lipid content of all samples was measured gravimetrically after evaporation of the organic solvent. Analyses were performed in triplicate.

The moisture and ash content were determined gravimetrically by drying the samples for 24 h at 105 °C for moisture and by combusting the samples in a muffle furnace (model N11, Nabertherm GmbH, Lilienthal/Bremen, Germany) at 550 °C for ash. Analyses were performed in triplicate.

### 2.3. Amino Acids and Peptide Profile

#### 2.3.1. Amino Acids

Total amino acid composition and free amino acids were determined as a subcontracted service with Ion Chromatography-UV-detector (IC-UV) using the ISO 13903:2005; EU 152/2009 standard method. With cysteine + cystine and methionine, the samples were oxidized with hydrogen peroxide and formic acid before the hydrolysis. The detection was carried out using post column derivatization with ninhydrin reagent at 440 and 570 nm. Tryptophan was determined using the ISO 13904:2016 standard method which included high-performance liquid chromatography (HPLC) and alkaline hydrolysis. Analyses were performed in triplicate.

#### 2.3.2. SDS-PAGE

Peptide weight was determined in reducing conditions by sodium dodecyl sulfide polyacrylamide gel electrophoresis (SDS-PAGE) according to Nisov et al. [8]. The commercial Criterion TGX (Tris-glycine extended) stain-free precast gel (4–20%, 30 µL 18-well, Bio-Rad, Hercules, CA, USA) was used for the analysis. The molecular weight (MW) of protein bands was estimated using Precision Plus Protein™ standards (10–250 K, Bio-Rad Lab., Inc., Hercules, CA, USA). Analyses were performed in duplicate.

#### 2.3.3. Molecular Weight (MW) Distribution by Using Size-Exclusion Chromatography

Size-exclusion chromatography (SEC) of commercial protein samples was performed by using Äkta pure 25 M chromatographic system (Cytiva Sweden AB, Uppsala, Sweden) equipped with ALIAS Bio autosampler (DURATEC Analysentechnik GmbH, Hockenheim, Germany) with Superdex 75 HR 10/30 column (GE Healthcare Life Sciences, Uppsala, Sweden). The analysis was performed according to Mäkinen et al. [15]. The total surface area of the chromatograms was integrated and separated into four MW ranges (>10,000, 1000–10,000, 200–1000, <200 Da), and the results were expressed as a percentage of the total area. Analyses were performed in quadruplicate.

### 2.4. Fatty Acid Composition 

Extracted oil (Section 2.2) was dissolved in chloroform and fatty acids were derivatized to methyl esters (FAME) using transesterification procedure with 2% (*v*/*v*) sulfuric acid in methanol [16]. A gas chromatograph (model 6890 N, Agilent Technologies, Santa Clara, CA, USA) fitted with a CP-Sil 88 column (100 m × 0.25 mm i.d., 0.2 μm, Agilent Technologies, Santa Clara, CA, USA) and flame ionization detector were used in quantifying FAME. A temperature gradient program was used in the oven, and hydrogen was used as the carrier gas (constant pressure 206.8 kPa; nominal initial flow rate 2.1 mL min^−1^ [16]). The fatty acid composition was calculated as weight percentages using theoretical response factors [17]. Analyses were performed in duplicate. 

### 2.5. Lipid Oxidation

The content of thiobarbituric acid reactive substances (TBARS) in the samples was measured according to Logren et al. [18]. Briefly, malondialdehyde (MDA) was released from proteins using alkaline hydrolysis after which acid precipitation was used to diminish the proteins. Supernatants were reacted with thiobarbituric acid (TBA) to form MDA-TBA adducts and Ultra-High-Performance Liquid Chromatography (UHPLC) analysis was used to measure the MDA-TBA adducts. Analyses were performed in quadruplicate.

Peroxide values (PVs) were determined from the extracted lipids using a ferric thiocyanate method as described by Lehtonen et al. [19]. PV could not be analyzed from samples COLL1 and HYDR2 due to their extremely low lipid contents. Results were calculated as meq/kg powder (“as is”) based on the lipid content of the powders. Analyses were performed in triplicate.

### 2.6. Determination of Functional and Sensory Properties

#### 2.6.1. Functional Properties

The nitrogen solubility was determined by dissolving 2.5 g of sample in milliQ water until 50.0 mL was reached. The 5% solution was mixed constantly for 30 min. The solution was centrifuged for 15 min at 10,000× *g* and the supernatant was collected for protein content measurements with the Dumas method (Section 2.2). Analyses were performed in triplicate.

Foaming capacity (FC) and foaming stability (FS) were determined from 5% (*w*/*w*) solution as described by Nisov et al. [20]. The foam was produced by Aerolatte (UK) by keeping the mixer 60 s still at the bottom of the measuring cylinder. The total volume (upper limit of the foam) and drainage (upper limit of the liquid part) were measured at time points of 0, 1, 5, 10, and 20 min. The results are presented as percentage of volume compared to the original volume. Analyses were performed in triplicate.

Heat induced gelation was evaluated according to Nisov et al. [8]. One milliliter of 15% (*w*/*w*) protein–water solutions was pipetted to a 2 mL tube, stirred, and heated at 98 °C for 20 min. Heated tubes were placed in a refrigerator overnight. The sample was classified as gel if the sample stayed at the bottom of the tube after tilting it upside down. Analyses were performed in triplicate.

Water holding capacity (WHC) was determined following the AOAC method 56-30.01. 

Fat binding capacity (FBC) of protein samples was determined by modified procedure of Cho et al. [21]. Briefly, 20 mg of protein sample was mixed with 1 mL of rapeseed oil and held at room temperature. The protein–oil mixture was stirred with a vortex mixer for 5 s at a time interval of 15 min. After 1 h, mixtures were centrifuged at 450× *g* for 20 min. The upper phase was removed, and the tube was drained for 30 min on a filter paper by tilting to a 45 angle. The FBC were calculated as the weight of the contents of the tube after draining divided by the weight of the dried protein sample and expressed as the g of oil/g of dried protein. Analyses were performed in triplicate.

#### 2.6.2. Sensory Properties

The color of the samples was measured with a colorimeter (Minolta Chroma meter, CR-200 Handheld, Osaka, Japan) by filling ⌀50 mm plastic petri dish with the sample. The results were reported in the CIE system as lightness (L*), red-green (a*), and yellow-blue (b*). Analyses were performed with five replicate measurements. The whiteness of the samples was calculated using the following equation [22]:(1)$$\mathrm{Whiteness}=100-\sqrt{{\left(100-L\right)}^{2}+a^{2}+b^{2}}$$

The sensory profile was determined with general descriptive analysis (GDA) [23]. Nine panelists from VTT Technical Research Centre of Finland’s trained sensory panel were recruited to evaluate the three-digit coded and randomized samples. The panel consisted of 6 females and 3 males. Informed consent was obtained from the panelists before the evaluation, where the possible allergens (fish and seafood) were disclosed, and the response confidentiality was assured. The panelists could withdraw themselves from the evaluation at any time without giving a reason. The microbiological safety of the tested products was assessed before the evaluations. The evaluations were performed as a taste-and-spit assay. Odor and appearance were analyzed from 5% (*w*/*w*) solutions of fish proteins in tap water whereas flavor and mouthfeel properties were analyzed from 1% (*w*/*w*) solutions. A volume of 30 mL of solution was given to the assessors. The samples were served in a randomized Latin square serving order. Appropriate concentrations of the protein solutions and preliminary lexicon for the samples were discussed with a subsection of the panelists and sensory experts. Altogether 15 sensory attributes were chosen for GDA (6 taste attributes, 4 odor attributes, and 5 mouthfeel, texture, and appearance attributes). Panelists were provided with reference samples for some attributes. The evaluated attributes and their respective descriptions and selected reference samples are listed in Appendix A. The intensities of the sensory attributes were evaluated in triplicate using an unstructured line scale with labeled endpoints ranging from no intensity (0) to high intensity (10) using EyeQuestion software version 5.0.8.5 (EyeQuestion Software, Logic8 B.V., Elst, The Netherlands). 

#### 2.6.3. Odor-Active Volatile Compounds

Volatile compounds were determined using HS-SPME-GC-MS/O. First, three samples (CONC3, HYDR1, and HYDR2) chosen by pretesting were analyzed with GC-MS-O with four trained assessors using detection frequency method for the identification of odor-active compounds. Then, relative concentration (RC) was determined for all samples by adding 1-butanol (1048 ng/vial) as an internal standard (ISTD), whose RC was set to 100, and comparing the compound area to ISTD’s compound area using the following equation:RC = (Compound area)/(ISTD’s compound area) × ISTD RC(2)

The odor-active compounds were collected from all identified volatile compounds with two methods: (1) collecting all odor-active compounds identified with GC-MS-O analysis and (2) additional compounds found in GC-MS analysis that have been determined as odor-active in prior literature [24,25,26,27].

The GC-MS/O analysis was performed using an Agilent 6890N gas chromatograph equipped with an Agilent 5973 mass spectrometer (Agilent Technologies, Santa Clara, CA, USA), with olfactometry ODP_4_ sniffing port, (Gerstel, Linthicum, Maryland, USA) and PAL3 autosampler system (RTC, CTC Analytics AG, Swingen, Switzerland). Samples were analyzed with a capillary column Vf-Wax (60 m × 0.25 mm i.d. × 0.5 µm, Agilent Technologies, Santa Clara, CA, USA). High-purity helium was applied as the carrier gas. GC eluent was split 1:1 with the detectors. With MS, the ion detection range was 25–450 m/z, the temperatures were 250 °C for the ion source and 150 °C for the quadrupole. The injector temperature was 250 °C. The temperature program for GC was modified based on Kakko et al. [28]: 40 °C held for 3 min, 40–150 °C at a rate of 12 °C/min, 150–240 °C at the rate of 10 °C/min and held at 240 °C for 5.8 min. The temperature for olfactometry and the transfer line was 220 °C. Samples were incubated (60 °C, 20 min) in 25% (*w*/*v*) water solution, and the volatile compounds were collected from the headspace of the 20 mL vial to SPME fiber (2 cm, DVB/CAR/PDMS, phase thickness 50/30 μm, Supelco Inc., Saint Loius, MO, USA). Samples were injected in splitless mode with a purge time of 8 min. Analyses were performed in triplicate.

The three samples that were first analyzed with the GC-MS-O were also analyzed with two assessors with an Agilent 6890 GC-FID/O (Agilent Technologies, Santa Clara, CA, USA) equipped with DB-5 (60 m × 0.25 mm i.d. × 1.0 µm, Agilent Technologies, Santa Clara, CA, USA) column to confirm the compound candidates. GC-FID/O used the same SPME- and GC parameters as GC-MS-O. FID temperature was 220 °C.

### 2.7. Statistical Analysis

The data were analyzed with analysis of variance (one-way ANOVA) with Tukey’s post hoc analysis using <5% as a limit for statistical significance. Kruskal–Wallis one-way ANOVA was used when the results were not normally distributed. Normality was tested with a Shapiro–Wilk test and by visual inspection of the distributions. With sensory evaluation, data were analyzed with a two-way mixed model analysis of variance (two-way ANOVA) with samples as the fixed factor and the assessors as a random factor. Statistical analysis was performed with IBM SPSS Statistics 28 for Windows (Version 27, IBM Corp., Armonk, NY, USA). The multivariate analyses were performed with Unscrambler X (version 10.5.1, Camo Software AS, Oslo, Norway). The panel performance in the sensory evaluation was analyzed with PanelCheck (version 1.4.2, Nofima, Tromsø, Norway).

## 3. Results and Discussion

### 3.1. Proximate Composition

The proximate composition of the samples is presented in Table 2. Overall, the concentrates contained more total lipids and ash compared to the hydrolyzed samples. All the samples were high in protein, but the hydrolysates had the highest protein content. With HYDR2 and COLL samples the ash content was remarkably low. With enzymatic hydrolysis, the soluble peptides are separated and collected from lipids and other insoluble matter but with heat treatment, the fat cells are merely ruptured, and the lipids are pressed out from minced raw material [6]. These differences in processing methods cause variation in the proximate composition. For the COLL1 sample, the conversion factor for protein is most likely too high, since the proximate composition is over 107% with the lipids, moisture, and ash taken into consideration. However, when we sum up the amino acids in the COLL1 sample (see Section 3.2.1 Amino acids), it sums up to 87.7 ± 16.65 g or protein/100 g of sample. This is likely closer estimation of the true protein content.

The part of fish used as a raw material in the fish proteins did not have an influence on the practical level the proximate composition of the samples, since the CONC2 (cod filleting by-products) and CONC3 (only cod backbones) had only minor differences between them. On the other hand, the results showed the effect of the raw material and the processing method on the lipid content of the samples. For example, in sample CONC1 (cod, saithe, and haddock), the lipid content was higher than in sample CONC2 (only cod). In addition, the HYDR2 (by-products of salmon filleting) had fewer lipids than HYDR1 (whole blue whiting), even though salmon is generally considered a slightly fattier fish than blue whiting. 

The total lipid content of sample CONC3 was 4.2% in dry matter, which is less than previously reported in the concentrate made from cod backbone (8.8–9.3%) [29,30] and more than reported with concentrates made from saithe and haddock fillets (0.5–0.6%) [31,32]. The difference may be due to the processing or lipid extraction methods. Saithe and haddock fillets [31,32] were analyzed with Soxhlet extraction, which is ineffective for extracting fish oil [33,34]. A study by Egerton et al. [35] found the lipid content of blue whiting hydrolysates to be between 0.0–3.0% of dry weight which is similar to the hydrolysate sample tested in this study (HYDR1, 0.4%). However, Slizyte et al. [36] reported a 2.2–6.2% lipid content of salmon backbone hydrolysates, which is different from the sample HYDR2 (0.1%) in this study. Despite the different findings, the methods used in both studies and the samples tested in this study are similar and the results are considered comparable. The lipid content of fish collagen from different species and fish parts is reported to be 0.2–1.2% [37,38,39] similar to the COLL1 sample (0.1%). Overall, the lipid content of concentrates is higher than hydrolysates and the collagen samples have a very low level of lipid content.

### 3.2. Amino Acid and Peptide Profile

#### 3.2.1. Amino Acids

Total amino acids of the commercial fish proteins are presented in Table 3. Amino acid content was the same across concentrates and hydrolysates with only minor statistically significant differences. The essential amino acid to non-essential amino acid ratios were high in concentrates and hydrolysates. Previously, pH-shift extracted protein isolates from two fishes were found to have higher essential to non-essential amino acid ratios compared to hydrolysates from same raw materials [40]. COLL1 on the other hand had major differences compared with the other proteins. Concentrates and hydrolysates contained adequate amounts of all essential amino acids. As expected, COLL1 lacked tryptophane and had low amounts of other essential amino acids as well. The only essential amino acid that COLL1 collagen had adequately was lysine. The lysine content was high also in concentrates and hydrolysates [41].

COLL1 had high amounts of glycine, proline, hydroxyproline, and alanine, which was in line with previous studies [42,43]. These amino acids were also high in HYDR2, which therefore likely contains collagen [43]. This could be due to the proportionally high amount of fish skin in salmon by-products compared to, e.g., whole blue whiting (HYDR1). CONC3 differed from CONC1 and CONC2 samples only with tryptophan and even with that, the difference was minimal. This is consistent with other studies. For example, Aspevik et al. [9] found that both backbones and heads have high nutritional values. Neither the fish species nor the part of the fish used seemed to affect to the nutritional quality of the protein in concentrates and hydrolysates. 

Free amino acids are presented in Table 4. Generally, the hydrolysates contained more free amino acids than concentrates or hydrolyzed collagen. Especially high free amino acid content can be found with sample HYDR1. Free amino acids were measured as an explaining factor for sensory properties and therefore they are discussed more in Section 3.6.

#### 3.2.2. Peptide Profile

The duplicate SDS-PAGE runs showed consistent molecular weight distribution results and therefore only one of the runs is presented in Figure 1. Hydrolyzed collagen often leads to peptides ranging from 3–5 kDa [44], as demonstrated previously with fish [43,45,46]. As expected, the collagen sample showed no bands except one very light band smaller than 10 kDa. Samples HYDR1 and HYDR2 showed molecular weight under 10 kDa apart from faint band with HYDR1 sample over 250 kDa. Nisov et al. [8] reported a similar faint band with hydrolyzed fish protein. However, the band over the 250 kDa mark did not appear in the HYDR2 sample which might be due to heavy filtration.

The protein concentrates showed a larger variation of different bands with an intense band at approximately 40 kDa referring to actin, which have been found previously with pH-shift method [8,47]. A band between 15 and 20 kDa indicated myosin light chains and the faint band between 150 and 250 kDa indicated myosin heavy chains. Samples CONC1 and CONC2 had degraded more than CONC3, which could be due to a different raw material. No significant differences were seen between fish species in CONC samples, which was expected since all the raw materials belonged to the same Gadidae family.

As can be seen from the SEC chromatograms (Figure 2), compounds with a molecular weight greater than 10 kDa comprised the most prominent fraction in the CONC samples (Figure 3). Both CONC1 and CONC2 had almost identical MW distribution profiles, while CONC3 had a lower proportion of compounds with molecular weight < 1 kDa compared to the other protein concentrates. It should be noted that the amount of dissolved protein obtained by Bio-Rad DC Protein Assay (Bio-Rad Laboratories, Hercules, CA, USA) for the fish concentrate samples was very low: 0.3 mg/mL, 0.4 mg/mL, and 0.6 mg/mL for CONC1, CONC2, and CONC3, respectively. Thus, obtained SEC profiles represent dissolved protein, not the total protein. However, it was found that the protein solubility for CONC samples was very low. Therefore, the obtained SEC results correspond well to the experiments performed in aqueous solutions.

The most prevalent fraction in both hydrolysate samples was 1–10 kDa. With HYDR1, compounds above 10 kDa were almost negligible, while with HYDR2 they were moderately present at about 15% of total protein (Figure 3). In the HYDR1, the proportion of compounds below 1 kDa, which correspond to peptides of fewer than ten amino acids or free amino acids, was the highest among the examined samples. Surprisingly, in HYDR2 their proportion was even lower than in CONC samples. The results show that in sample HYDR1 the enzymatic treatment effectively hydrolyzed the fish proteins into peptides, having MW < 1 kDa. In general, bioactive peptides that are suitable for use in therapeutic foods primarily have a molecular weight of less than 1 kDa [48].

Although the proportion of > 10 kDa in the COLL1 sample was of the same order of magnitude as in the CONC samples, the SEC profile was different. It lacked the major peak near the void volume typical of CONC samples (Figure 3). In addition, there were practically no compounds smaller than 1 kDa in the COLL1 samples.

### 3.3. Fatty Acid Composition

Fish oil is widely regarded as an important source of long-chain polyunsaturated fatty acids (PUFA), particularly eicosapentaenoic acid (EPA, 20:5n-3) and docosahexaenoic acid (DHA, 22:6n-3). Eating fish has many health benefits that have been linked to n-3 fatty acids. For example, clinical trials have shown that fish oils prevent arterial hypertension, type 2 diabetes, and memory impairment in older people [49]. Indeed, the European Food Safety Authority (EFSA) has approved several health claims related to EPA and DHA, listed in Välimaa et al. [50]. The fatty acid compositions of the commercial protein concentrates are shown in Table 5. The fatty acid compositions of HYDR1, HYDR2, and COLL1 could not be determined due to analytical problems. However, the total amount of lipids in these samples were very low (0.4%, 0.1%, and 0.1%, respectively), making the composition of the fatty acids an unimportant factor.

All protein concentrates were good sources of n-3 fatty acids as in CONC1, CONC2, and CONC3 the proportion of n-3 fatty acids was 29.2%, 34.2%, and 40.9% of the total amount of fatty acids, respectively. No statistically significant differences were found between the samples. Interestingly, the amount of DHA seemed to be inversely proportional to the oxidation of the concentrates. According to Falch et al. [51], the DHA content of cod, saithe, and haddock by-products was of the same order of magnitude (except for saithe caught in winter it was lower). Since CONC1 is made from the by-products of cod, saithe, and haddock, and CONC2 and CONC3 from the by-products of cod filleting and backbone, respectively, the DHA concentrations of the samples could be expected to be quite similar. Therefore, the lower DHA content of CONC1 may be due to the degradation of DHA caused by oxidation. 

### 3.4. Lipid Oxidation

Primary lipid oxidation in fish protein concentrates and hydrolysates were assessed by quantification of PVs (Figure 4). Due to the extremely low lipid content of COLL1 and HYDR2 (0.1% and 0.1%, respectively), no PVs were obtained for these two samples. Of the rest of the samples, HYDR1 had by far the lowest PV. The concentrates (CONC1, CONC2, and CONC3) had higher PVs compared to the hydrolysate, but considerably lower than what was previously observed for roach and Baltic herring protein isolates (3–25 meq/kg sample) [40], and lower than for Nile tilapia muscle during frozen storage (1.7–4 meq/kg sample) [52]. CONC1 (by-products of cod, saithe, and haddock) had a significantly higher PV (meq/kg powder) compared to CONC2 and CONC3 from cod, which was likely due to its higher lipid content (Table 2). However, due to the fast decomposition of hydroperoxides, PV should be considered together with secondary lipid oxidation products.

Lipid oxidation in the fish protein concentrates and hydrolysates was assessed further by measuring the TBARS as MDA content (Figure 5). MDA is formed as a secondary product in the lipid oxidation [53]. Among the samples, HYDR1 showed the highest MDA content whereas COLL1 had the lowest (Figure 5). The higher MDA concentration of the HYDR1 in comparison to the COLL1 could be partly because the HYDR1 sample has a higher total lipid content (Table 2), but it is unlikely that this is the major factor. This is because the CONC samples, which contained even more lipids (fats) than the HYDR1 sample, had a lower concentration of MDA. The results suggest that fish species is one putative factor affecting the lipid oxidation level. Among the samples, HYDR1 is the only one prepared from blue whiting. Thereby, the results indicated that blue whiting fatty acids may be more susceptible to oxidation in comparison to the other samples. The MDA content of HYDR1 was similar to protein hydrolysate prepared from saithe fillets with alkali aided hydrolysis (90.5 nmol/g) [32]. In addition, higher MDA content compared to this study have been reported for example, for pickled Baltic herring fillets [18] and marinated Atlantic herring [54].

### 3.5. Functional Properties

The functional properties of fish proteins are presented in Figure 6. Protein solubility in water (Figure 6A) results follow the consensus that protein hydrolysates are extremely soluble due their low molecular mass and high concentration of ionizable groups [55] whereas protein concentrates are less soluble due to protein denaturation caused by heating during processing [56,57]. However, if only the middle soluble part of heated fish-water suspension is collected in protein concentrate production, the proteins have higher solubility ranging from 63.4% to 87% as demonstrated by Sathivel et al. [58]. Solubility is a crucial functionality in proteins as it affects proteins’ WHC, gelation, foaming, and emulsification abilities and thus the usability of the product in foodstuffs [55,59].

FBC is an important function in a protein since the fat-binding ability affects the mouthfeel and texture of the food [60]. FBCs were low in all samples (Figure 6B) compared to previous literature with hydrolysates and concentrates [58,61]. In comparison, the hydrolyzed samples had higher FBC than concentrated samples, since with enzymatic hydrolysis more hydrophobic peptides are available [55]. 

Protein ingredients need to possess some water holding capacity to function in a food matrix since it fundamentally affects the texture properties such as mouthfeel and juiciness [60]. WHC of the food-grade samples is presented in Figure 6C. The WHC of CONC samples was good compared to plant and soy protein and very similar to whey protein [60]. Fish meals have been reported to have even higher WHC [56]. Hydrolyzed samples had poor WHC compared to previous research [62]. Hydrolysates were expected to have higher WHC due to hydrolysates’ low molecular weight and high protein–water interactions, but the solubilization of hydrophilic protein groups might affect WHC [55].

Foaming properties of proteins affects the body, smoothness, and lightness of the food [60]. Hydrolyzed samples showed better FC than the protein concentrates (Figure 6D). Enzymatic hydrolysis has been shown to improve FC in various matrices [55]. The FC of the hydrolyzed samples was over 200%, which is quite high compared to previous research [8,63]. However, good FC indicates a suitable degree of hydrolysis and high solubility [60]. High solubility and hydrophobic amino acids have been associated with better FS [55]. The effect of hydrophobic amino acids does not seem to be significant with fish proteins, since HYDR1 has the most hydrophobic amino acids but has the poorest FS (Figure 6E).

Gelling properties of proteins affects the texture of food, mainly juiciness, body, and mouthfeel [60]. Unfortunately, none of the samples showed any gelation properties at 15% concentration. To form a rigid gel network, proteins need to form covalent bonds and aggregate between denaturized protein strands [55]. The lack of gelling properties can be a result of a too-high degree of hydrolysis [8]. With concentrates, the low solubility and higher fat content can be factors behind lack of gelation properties. However, surimi or isolates produced with pH-shift can form gel networks usable in food solutions [64,65].

To sum up, regarding the functionality of the proteins, the usability of fish proteins mainly depends on the production method. Based on solubility and foaming properties, fish protein hydrolysates (HYDR and COLL samples) could be used in liquid-based foods or beverages in addition to solid foods [60]. On the contrary, with good WHC, fish protein concentrates (CONC samples) could be added to solid foods to add nutritional value to, e.g., pasta, but due to the low solubility, the possibly gritty mouthfeel could decrease the pleasantness of the fortified food.

### 3.6. Sensory Evaluation, Odor-Active Compounds, and Other Explanatory Factors

The color of the samples is presented in Table 6. Nisov et al. [8] found hydrolyzed herring and roach proteins whiter than pH-shifted alternatives. However, Sathivel et al. [58] found even higher whiteness values when studying herring and arrowtooth flounder (*Atheresthes stomias*) protein powder produced by heat treatment. Sathivel et al. [66] determined that lipid oxidation plays a role in yellow discoloration. This negative correlation was also seen in the present study: HYDR1 had the highest TBARS values (100.5 nmol MDA/g) and the lowest whiteness value of all the hydrolyzed samples (66.4). In addition, HYDR1 had the highest yellow-blue value (27.0) of all samples. CONC1 had a significantly higher peroxide value (15.3 meq/kg oil) and TBARS (33.1 nmol MDA/g) values of all CONC samples and the lowest whiteness value (62.5) of all the samples. The appearance of the fish proteins could possibly be improved if lipid oxidation is prevented.

The intensities of the sensory properties are found in Table 7, and the spider plot of mean values is found in the Appendix A. The sensory evaluation clustered the proteins according to their processing method, which has been observed previously by Nisov et al. [8]. However, all the proteins were equally intense in flavor and odor except for COLL1. The PLS correlation loadings plot of explanatory factors for the sensory properties of fish proteins is presented in Figure 7. Umami taste was strongly associated with HYDR1, which could be due to relatively large amounts of free glutamic acid and aspartic acid compared to other samples [67,68]. Additionally, complex flavor formation has been associated before with a large number of free amino acids [67], which could be affecting the taste of HYDR samples.

In GDA, hydrolysates were found to be more bitter and more metallic than the concentrates. The results were in accordance with previous studies that demonstrated bitterness to be the main sensory challenge with fish hydrolysates [9,68,69,70]. The peptide size analysis showed that the HYDR 1-2 samples consisted mostly of peptides under 10 kDa, which have previously been associated with a bitter taste [68]. Peptides with exposed hydrophobic side chains are associated with a bitter taste [71], and with smaller peptides, more side chains are exposed leading to a more bitter taste. Contrary to the results of Halldorsdottir et al. [70], the secondary lipid oxidation products did not seem to affect the intensity of bitterness in this study: HYDR2 was perceived as more bitter than HYDR1, which had significantly higher TBARS values. Hydrophobic amino acids (Trp, Phe, Val, Leu, Iso, Try) are a key factor affecting the bitterness of the protein [72]. However, significant differences could not be found in the amino acid analysis between concentrates and hydrolysates, but the free amino acid analysis showed a clear and significant increase of hydrophobic amino acids in hydrolysates compared to concentrates. These free amino acids combined with peptide size are likely the key factors affecting the bitterness of the hydrolysates. A diverse peptide size distribution was found in CONC samples (see Section 3.2.2. Peptide profile), which has been associated with complex taste sensations [71] that can be affecting the formation of fishy flavor and flavor intensity. The nearly tasteless COLL sample had the most monotonous peptide size distribution.

A total of 33 compounds were detected with GC-MS-O, from which 20 could be identified. After including the additional compounds typically mentioned in the literature (as described in Section 2.6.3) via the GC-MS, 46 different odor-active compounds likely affecting the odor and taste of the fish proteins were found. Out of these 46 compounds, 33 could be identified. It is important to mention that the compounds detected through olfactometry play a larger role in creating the odor of fish protein and therefore are given more emphasis in the discussion. Detection frequencies (DF) of compounds detected with GC-MS-O are found in Appendix A. The compounds and their relative concentrations are presented in Table 8. Four compounds could not be quantified with GC-MS, so with those compounds and with unidentified compounds, only detection frequency is shown. Hydrolyzed samples had more unidentified odor-active compounds suggesting the need for further information on compounds affecting fish hydrolysates’ odor, such as brothy odor. The odor and flavor properties identified with GDA correlated well with the odor-active compounds.

The PLS correlation loading plot found in Appendix A showed a correlation between odor and taste intensity and methional, trimethylamine (TMA), and 3-methylbutanal. Methional has such a low odor threshold that it could be a significant compound affecting the odor and taste intensity. Trimethylamine has been determined to be one of the most important fishy odor sources in fish products [73]. All the samples, apart from COLL1, contained trimethylamine. The prevention of TMA formation is crucial for producing fish proteins with mild odor and taste. Recently, Goris et al. [74] demonstrated enzyme-mediated conversion to TMAO as a working approach to reduce the TMA content in fish proteins. 3-Methylbutanal is formed by the degradation of isoleucine and the concentration increases as the degradation of fish progresses [75]. 3-Methylbutanal has previously been found, for example, in surimi samples [76] and Baltic herring [28]. This sour-bread-smelling compound could be affecting the overall odor and taste intensity in processed fish products.

The concentrates were found to be fishier and more sea or seaweed-like than the hydrolysates, which have been reported to be an issue with all fish proteins regardless of the processing method [8,22,32,58]. In addition to TMA, fishiness of the sample has largely been associated with secondary lipid oxidation products [32,68,77]. However, fishiness did not seem to be associated with TBARS, since the values were the same between, e.g., samples HYDR2 and CONC3, but the fishy odor and taste intensity differed largely between the samples. Association between fishiness and lipid oxidation was found from the odor-active volatile compounds. Fish protein concentrates had the most abundant amounts of lipid oxidation derived volatile compounds, such as 2,3-butanedione, (*Z*)-4-heptenal, octanal, heptanal, and hexanal. These compounds have been associated with fishy odor and taste development [73]. 2,3-Butanedione, (*Z*)-4-heptenal, and hexanal were previously reported as significant odorants in pH-shifted protein isolate from Baltic herring [28]. 

COLL1 was the whitest, and mildest in taste and odor, which are desired properties of fish protein. The COLL1 was positioned on the opposite side of the ellipse from odor-active compounds such as TMA, (*Z*)-4-heptenal, methional, and 3-methylbutanal (Figure 7). This indicates that this study successfully identified the odor-active compounds of fish commercial fish proteins. However, rotting-cabbage-smelling methanethiol, which was detected also with GC-MS-O, was associated with the COLL1 sample, which could be affecting its mild odor, and thus the odor activity of methanethiol should be taken into consideration. Methanethiol has been previously found in fish sauce [78]. Methanethiol, hydroxyacetone, and 2-heptanone were found in fish proteins for the first time in this study.

To produce mild-tasting and -smelling fish protein products, the lipid oxidation and degradation of raw material should be assessed carefully. Lipid oxidation could be prevented, e.g., with the addition of antioxidants [52]. To limit the degradation of raw materials, appropriate storage conditions and quick processing after capture is essential. Additionally, the storage stability of final products throughout the shelf-life should be assessed carefully.

## 4. Conclusions

The processing method has been shown to fundamentally affect the functional properties of fish proteins and thus, the usability in food solutions. In this study, raw material had some effect on the proximate composition of the proteins, but not on the chemical or sensory properties. Generally, protein hydrolysates had better functional properties compared to concentrates, therefore making them the most potential option for food ingredients. However, concentrates had a better amino acid composition, higher amount of lipids, and better fatty acid quality. This leads to better nutritional value in fish concentrates, if used as food ingredients. However, the sensory properties, mainly fishiness and bitterness, are potentially limiting the food-grade fish protein markets. These challenges are likely caused by lipid oxidation and raw material degradation, and limiting those with, e.g., adding antioxidants during processing or low oxygen processing conditions would likely lead to milder tasting and smelling product more suitable for human consumption.

## Figures and Tables

**Figure 1 foods-12-00966-f001:**
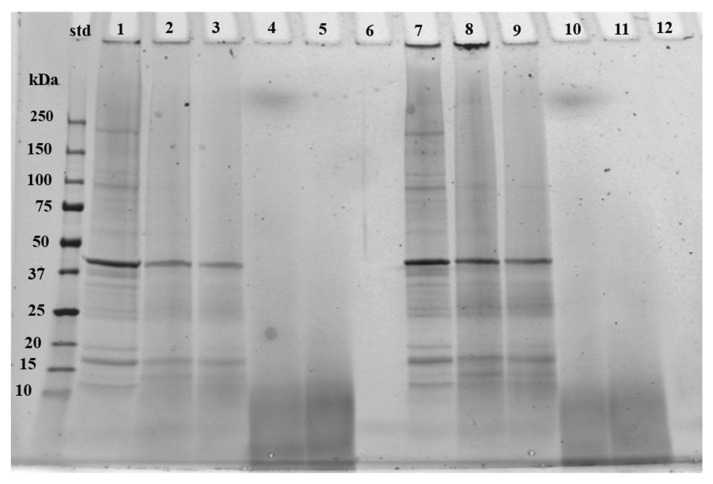
The reducing SDS-PAGE image of commercial fish protein concentrates, hydrolysates, and hydrolyzed collagen. To lines 1–6 was pipetted 20 µL of solution and to lines 7–12 25 µL of solution. Line numbers represent samples: 1 and 7 = CONC3, 2 and 8 = CONC1, 3 and 9 = CONC2, 4 and 10 = HYDR1, 5 and 11 = HYDR2, 6 and 12 = COLL1. Fish protein concentrates = CONC1–3, fish protein hydrolysates = HYDR1–2, hydrolyzed fish collagen = COLL1. Standard is abbreviated as std.

**Figure 2 foods-12-00966-f002:**
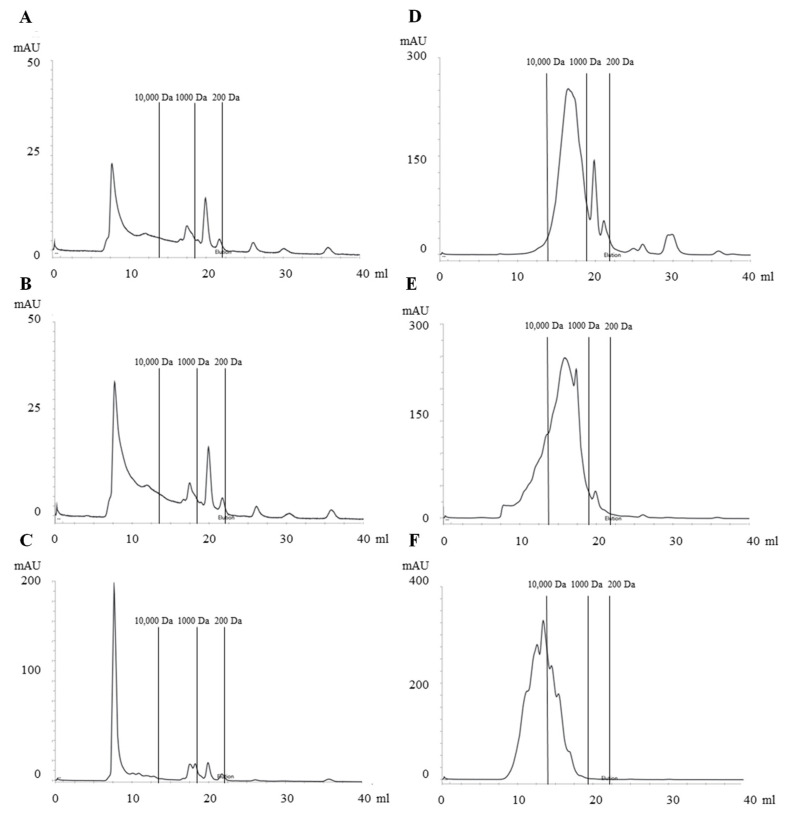
Size exclusion chromatograms (SEC) of commercial fish protein concentrates, hydrolysates, and hydrolyzed collagen. (**A**) CONC1, (**B**) CONC2, (**C**) CONC3, (**D**) HYDR1, (**E**) HYDR2, and (**F**) COLL1. Fish protein concentrates = CONC1–3, fish protein hydrolysates = HYDR1–2, hydrolyzed fish collagen = COLL1.

**Figure 3 foods-12-00966-f003:**
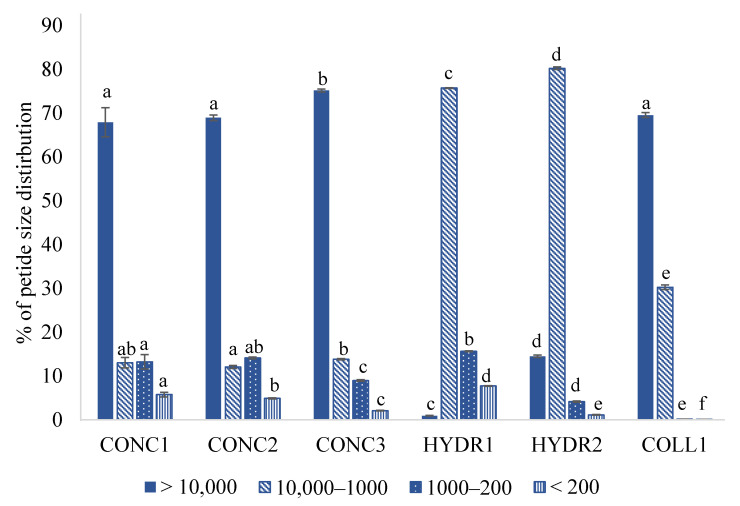
Proportion (%) of molecular weight (MW) distributions of commercial fish protein concentrates, hydrolysates, and hydrolyzed fish protein. Fish protein concentrates = CONC1–3, fish protein hydrolysates = HYDR1–2, hydrolyzed fish collagen = COLL1. Different letters within the same molecular weight grade indicate a statistically significant difference with ANOVA post hoc pair comparison (*p* < 0.05).

**Figure 4 foods-12-00966-f004:**
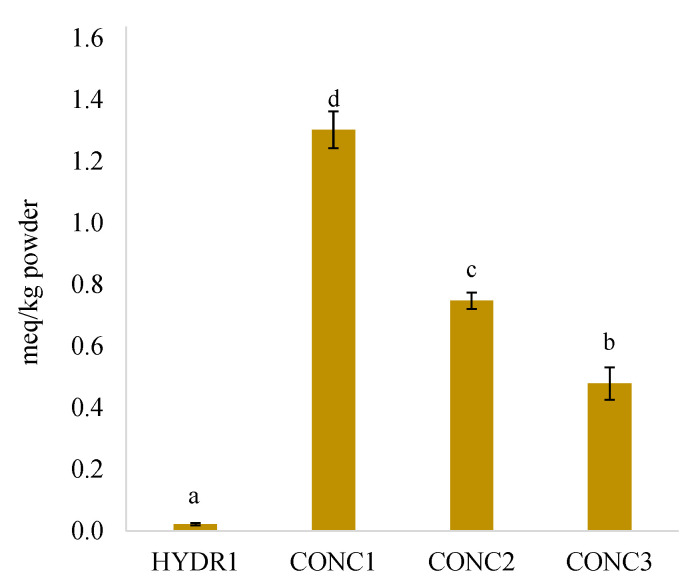
Peroxide values (meq/kg protein powder, “as is”) in three commercial fish protein concentrate (CONC1–3) and a hydrolysate (HYDR1). Different letters indicate a statistically significant difference with ANOVA post hoc pair comparison (*p* < 0.05).

**Figure 5 foods-12-00966-f005:**
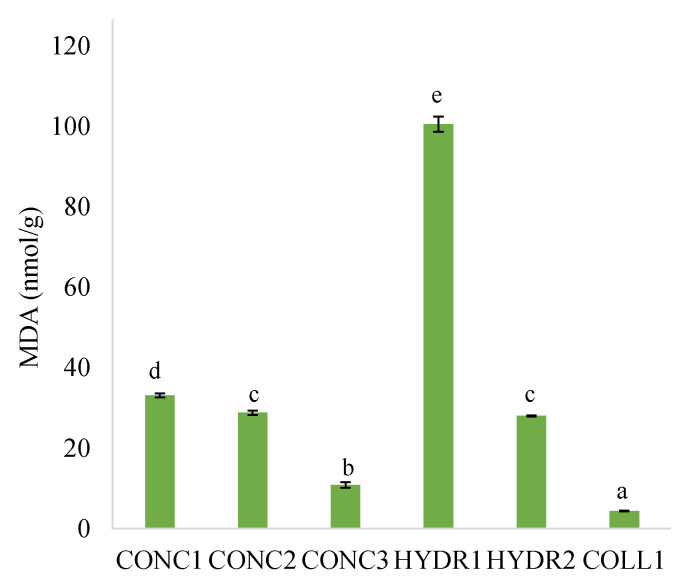
Malondialdehyde (MDA) contents (nmol MDA/g powder “as is”) of commercial fish protein concentrates, hydrolysates, and hydrolyzed collagen. Fish protein concentrates = CONC1–3, fish protein hydrolysates = HYDR1–2, hydrolyzed fish collagen = COLL1. Different letters indicate a statistically significant difference with ANOVA post hoc pair comparison (*p* < 0.05).

**Figure 6 foods-12-00966-f006:**
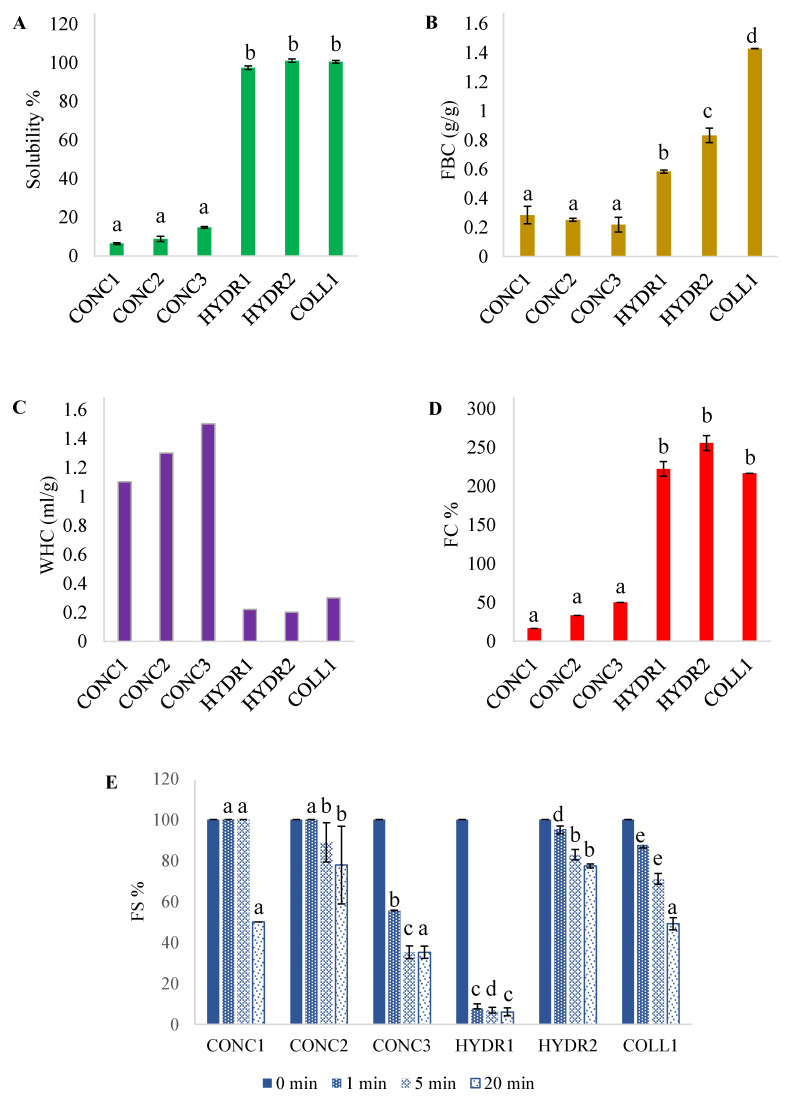
Functional properties of commercial fish protein concentrates, hydrolysates, and hydrolyzed collagen. (**A**) Protein solubility, (**B**) water holding capacity (WHC), (**C**) fat binding capacity (FBC), (**D**) foaming capacity (FC), and (**E**) foaming stability (FS). In figure (**E**), the results are compared by timepoints between the samples. Different letters indicate a statistically significant difference with ANOVA post hoc pair comparison (*p* < 0.05). Fish protein concentrates = CONC1–3, fish protein hydrolysates = HYDR1–2, hydrolyzed fish collagen = COLL1.

**Figure 7 foods-12-00966-f007:**
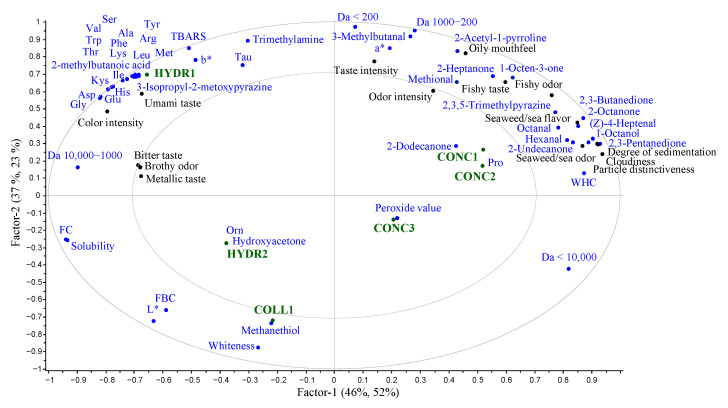
The Partial Least Squares (PLS) correlation loading of commercial fish protein concentrates, hydrolysates, and hydrolyzed collagen, where the free amino acids, lipid oxidation, peptide profile, odor-active volatile compounds identified with GC-MS-O, color, solubility, water holding capacity (WHC), fat binding capacity (FBC), and foaming capacity (FC) are the predictors (blue) of sensory properties (black). Fish protein concentrates = CONC1–3, fish protein hydrolysates = HYDR1–2, hydrolyzed fish collagen = COLL1.

**Table 1 foods-12-00966-t001:** The processing methods and raw materials of the commercial fish protein samples.

Sample	Raw Material	Part Used
CONC1	Cod (*Gadus morhua*), Saithe (*Pollachius virens*), and Haddock (*Melanogrammus aeglefinus*)	Fish filleting by-products
CONC2	Cod (*Gadus morhua*)	Fish filleting by-products
CONC3	Cod (*Gadus morhua*)	Cod backbones
HYDR1	Blue whiting (*Micromesistius poutassou*)	Whole fish
HYDR2	Salmon (*Salmo salar*)	Fish filleting by-products
COLL1	NA	NA

NA = Information not available.

**Table 2 foods-12-00966-t002:** Proximate composition of the commercial fish protein concentrates, hydrolysates, and hydrolyzed collagen with mean and standard deviation of three replicate measurements. Different letters within the same row indicate a statistically significant difference with ANOVA post hoc pair comparison (*p* < 0.05).

Sample	Protein (%)	Total Lipids (%)	Moisture (%)	Ash (%)
CONC1	63.0 ± 0.00 ^a^	8.5 ± 0.07 ^a^	4.2 ± 0.11 ^a^	21.4 ± 0.04 ^a^
CONC2	70.8 ± 0.00 ^b^	4.8 ± 0.08 ^b^	4.0 ± 0.21 ^b^	22.5 ± 0.03 ^b^
CONC3	69.3 ± 0.00 ^c^	4.1 ± 0.04 ^c^	5.3 ± 0.81 ^ac^	19.1 ± 0.02 ^c^
HYDR1	82.9 ± 0.00 ^d^	0.4 ± 0.01 ^d^	4.1 ± 0.03 ^ac^	9.8 ± 0.40 ^d^
HYDR2	93.4 ± 0.00 ^e^	0.1 ± 0.01 ^e^	3.0 ± 0.01 ^d^	2.7 ± 0.01 ^e^
COLL1	99.3 ± 0.00 ^f^	0.1 ± 0.02 ^e^	7.8 ± 0.01 ^d^	0.2 ± 0.01 ^f^

Fish protein concentrates = CONC1–3, fish protein hydrolysates = HYDR1–2, hydrolyzed fish collagen = COLL1.

**Table 3 foods-12-00966-t003:** Total amino acids (mg/g protein) in commercial fish protein concentrates, hydrolysates, and hydrolyzed collagen with mean and standard deviation of three replicate measurements. The amounts are compared to the daily requirement (mg/kg) [41]. Different letters within the same row indicate a statistically significant difference with ANOVA post hoc pair comparison (*p* < 0.05).

Essential	CONC1	CONC2	CONC3	HYDR1	HYDR2	COLL1	[41]
His	23.8 ± 3.33 ^a^	23.4 ± 3.28 ^a^	19.6 ± 2.75 ^a^	18.7 ± 2.62 ^a^	23 ± 3.22 ^a^	7.6 ± 1.06 ^b^	15
Ile	47.6 ± 6.67 ^a^	46.5 ± 6.51 ^a^	38.1 ± 5.34 ^ab^	40.8 ± 5.70 ^ab^	32.5 ± 4.54 ^b^	16.4 ± 2.30 ^c^	30
Leu	83.0 ± 11.62 ^a^	81.8 ± 11.46 ^a^	69.6 ± 9.74 ^a^	76.2 ± 10.67 ^a^	57.3 ± 8.02 ^a^	32.6 ± 4.57 ^b^	59
Lys	83.8 ± 11.73 ^a^	87.3 ± 12.23 ^a^	75.1 ± 10.52 ^a^	85.9 ± 12.02 ^a^	81.2 ± 11.36 ^a^	50.5 ± 7.06 ^b^	45
Met	32.4 ± 4.54 ^a^	32.8 ± 4.59 ^a^	28.5 ± 4.00 ^ab^	26 ± 3.64 ^a^	26.8 ± 3.75 ^a^	14.6 ± 2.04 ^b^	22 ^1^
Phe	22.8 ± 6.22 ^a^	43.3 ± 6.06 ^a^	36.6 ± 5.12 ^a^	37.9 ± 5.31 ^a^	31.7 ± 4.43 ^ab^	23.0 ± 3.21 ^b^	38 ^2^
Thr	50.5 ± 7.06 ^a^	47.9 ± 6.71 ^a^	41.2 ± 5.77 ^ab^	41.7 ± 5.84 ^ab^	43.8 ± 6.14 ^ab^	30.9 ± 4.33 ^b^	23
Trp	12 ± 1.21 ^a^	11.7 ± 1.17 ^a^	9.2 ± 0.92 ^b^	7.6 ± 0.76 ^b^	5 ± 0.49 ^c^	<0.01 ^d^	6
Val	28.5 ± 7.67 ^a^	53.4 ± 7.48 ^a^	44.9 ± 6.28 ^a^	47.3 ± 6.62 ^a^	41.8 ± 5.85 ^ab^	28.7 ± 4.02 ^b^	39
Non-Essential							
Ala	67.1 ± 9.40 ^a^	67.6 ± 9.48 ^a^	65.8 ± 9.21 ^a^	68.7 ± 13.58 ^a^	86.1 ± 12.06 ^a^	121.9 ± 17.06 ^b^	
Arg	71.4 ± 10.00	72.6 ± 10.16	67.6 ± 9.47	64.8 ± 9.07	79.4 ± 11.11	94.1 ± 13.17	
Asp	109.2 ± 15.29 ^a^	107.8 ± 15.10 ^a^	95.4 ± 13.36 ^ab^	102.7 ± 14.38 ^a^	106 ± 14.84 ^a^	65.6 ± 9.17 ^b^	
Glu	146 ± 20.44	154.5 ± 21.62	141.2 ± 19.77	154.3 ± 21.61	161.7 ± 22.64	110.8 ± 15.5	
Gly	79.5 ± 11.13 ^a^	77.7 ± 10.87 ^a^	94.9 ± 13.29 ^a^	70.9 ± 9.92 ^a^	161.7 ± 22.64 ^b^	287 ± 40.1 ^c^	
Hyp	9.9 ± 1.97 ^a^	10.4 ± 2.09 ^a^	18.6 ± 3.73 ^a^	11.6 ± 2.31 ^a^	43.9 ± 8.78 ^b^	96.8 ± 19.36 ^c^	
Orn	<0.05	<0.05	<0.05	<0.05	<0.05	<0.05	
Pro	50.0 ± 7.00 ^ab^	49.2 ± 6.89 ^ab^	50.8 ± 7.12 ^ab^	41.8 ± 5.86 ^a^	73.3 ± 10.26 ^b^	131.9 ± 18.47 ^c^	
Ser	53.3 ± 7.46	50.9 ± 7.12	49.7 ± 6.96	43.8 ± 6.12	51.1 ± 7.15	40.2 ± 4.43	
Tyr	36.2 ± 5.06 ^a^	36.1 ± 5.05 ^a^	29.4 ± 4.11 ^ab^	29.4 ± 4.12 ^ab^	22 ± 3.07 ^b^	3.5 ± 0.49 ^c^	
Cys	11.1 ± 1.56 ^a^	10.3 ± 1.44 ^a^	9.7 ± 1.26 ^a^	9.2 ± 1.29 ^a^	5.8 ± 0.80 ^b^	0.3 ± 0.04 ^c^	
Essential:non-essential	0.61 ± 0.01 ^a^	0.67 ± 0.00 ^a^	0.58 ± 0.00 ^a^	0.64 ± 0.00 ^a^	0.43 ± 0.00 ^a^	0.21 ± 0.00 ^b^	

Fish protein concentrates = CONC1–3, fish protein hydrolysates = HYDR1–2, hydrolyzed fish collagen = COLL1; ^1^ Methionine + cysteine (16 + 6); ^2^ Phenylalanine + tyrosine.

**Table 4 foods-12-00966-t004:** Free amino acids (µg/g) in commercial fish protein concentrates, hydrolysates, and hydrolyzed collagen with mean and standard deviation of three replicate measurements. Different letters within the same row indicate a statistically significant difference with ANOVA post hoc pair comparison (*p* < 0.05).

Essential	CONC1	CONC2	CONC3	HYDR1	HYDR2	COLL1
His	<0.18	<0.18	<0.18	2.12 ± 0.34 ^a^	0.63 ± 0.1 ^b^	<0.18
Ile	<0.18	<0.18	<0.18	7.4 ± 0.74 ^a^	0.61 ± 0.06 ^b^	<0.18
Leu	0.19 ± 0.02 ^a^	<0.08	0.1 ± 0.01 ^a^	26 ± 2.08 ^b^	2.46 ± 0.2 ^c^	<0.08
Lys	0.3 ± 0.05 ^a^	0.13 ± 0.02 ^a^	0.14 ± 0.02 ^a^	9.31 ± 1.49 ^b^	1.15 ± 0.18 ^a^	<0.07
Met	<0.08	<0.08	<0.08	7.86 ± 0.63 ^a^	0.59 ± 0.05 ^b^	<0.08
Phe	<0.16	<0.16	<0.16	12.7 ± 1.78 ^a^	0.77 ± 0.11 ^b^	<0.16
Thr	<0.03	0.07 ± 0.01 ^a^	<0.03	5.51 ± 0.55 ^b^	0.5 ± 0.05 ^a^	<0.03
Trp	<0.1	<0.1	<0.1	1.94 ± 0.19 ^a^	0.16 ± 0.02 ^b^	<0.1
Val	0.1 ± 0.01 ^a^	<0.08	0.09 ± 0.01 ^a^	10.5 ± 1.26 ^b^	1.12 ± 0.13 ^a^	<0.08
Non-Essential						
Ala	0.37 ± 0.03 ^a^	0.24 ± 0.02 ^a^	0.47 ± 0.04 ^a^	9.43 ± 0.75 ^b^	1.46 ± 0.12 ^c^	0.14 ± 0.01 ^a^
Arg	0.25 ± 0.03 ^a^	0.08 ± 0.01 ^a^	0.05 ± 0.01 ^a^	14.1 ± 1.41 ^b^	1.09 ± 0.11 ^a^	< 0.05
Asp	<0.09	<0.09	<0.09	2.47 ± 0.22 ^a^	0.79 ± 0.07 ^b^	0.2 ± 0.02 ^c^
Glu	<0.11	<0.11	<0.11	6.26 ± 0.44 ^a^	2.14 ± 0.15 ^b^	<0.11
Gly	0.31 ± 0.03 ^ab^	0.18 ± 0.02 ^a^	0.3 ± 0.03 ^ab^	1.74 ± 0.16 ^c^	0.66 ± 0.06 ^d^	0.43 ± 0.04 ^b^
Tau	3.05 ± 0.31 ^a^	1.71 ± 0.17 ^b^	4.41 ± 0.44 ^c^	5.9 ± 0.59 ^d^	3.02 ± 0.3 ^a^	<0.09
Orn	<0.1	<0.1	<0.1	<0.1	0.18 ± 0.02	<0.1
Pro	<0.1	0.14 ± 0.01	<0.1	<0.1	<0.1	<0.1
Ser	<0.08	<0.08	<0.08	4.52 ± 0.5 ^a^	0.68 ± 0.07 ^b^	<0.08
Tyr	<0.12	<0.12	<0.12	7.03 ± 1.27 ^a^	0.6 ± 0.11 ^b^	<0.12
Cys	<0.07	0.11 ± 0.02 ^a^	<0.07	5.71 ± 0.91 ^b^	1.74 ± 0.28 ^c^	<0.07

Fish protein concentrates = CONC1–3, fish protein hydrolysates = HYDR1–2, hydrolyzed fish collagen = COLL1.

**Table 5 foods-12-00966-t005:** Fatty acid composition (percent of total fatty acids) of the commercial fish protein concentrates (CONC1–3) with mean and standard deviation of two replicate measurements. There were no statistically significant differences between the samples with Kruskal–Wallis one-way analysis of variance.

Fatty Acid	CONC1	CONC2	CONC3
C14:0	3.1 ± 0.0	2.6 ± 0.0	1.5 ± 0.0
C16:0	15.4 ± 0.1	17.8 ± 0.0	17.8 ± 0.1
C18:0	3.5 ± 0.0	3.5 ± 0.0	4.2 ± 0.0
Total SFA	23.6 ± 0.2	25.2 ± 0.0	24.7 ± 0.1
C16:1 n-7	6.0 ± 0.0	3.8 ± 0.0	2.2 ± 0.0
C18:1 n-9	13.4 ± 0.1	11.5 ± 0.0	11.4 ± 0.0
C18:1 n-7	5.4 ± 0.0	4.1 ± 0.0	3.6 ± 0.0
C20:1 n-9	5.9 ± 0.0	6.1 ± 0.0	3.9 ± 0.0
C22:1 n-11	2.8 ± 0.0	3.1 ± 0.0	1.2 ± 0.0
C24:1 n-9	0.9 ± 0.0	1.2 ± 0.0	1.5 ± 0.0
Total MUFA	41.5 ± 0.1	35.8 ± 0.1	29.4 ± 0.2
C18:3 n-3	0.4 ± 0.0	0.4 ± 0.0	0.3 ± 0.0
C18:4 n-3	1.8 ± 0.0	1.0 ± 0.0	0.5 ± 0.0
C20:5 n-3	11.4 ± 0.1	11.4 ± 0.0	12.6 ± 0.0
C22:5 n-3	1.1 ± 0.0	1.0 ± 0.0	1.0 ± 0.0
C22:6 n-3	13.0 ± 0.1	19.6 ± 0.0	25.8 ± 0.2
Total n-3	29.2 ± 0.3	34.2 ± 0.1	40.9 ± 0.3
C18:2 n-6	1.2 ± 0.0	1.2 ± 0.0	0.9 ± 0.0
C20:4 n-6	1.6 ± 0.0	1.7 ± 0.0	2.3 ± 0.0
Total n-6	3.7 ± 0.0	3.6 ± 0.0	3.8 ± 0.0
Total PUFA	34.5 ± 0.3	38.6 ± 0.1	45.3 ± 0.2

SFA = saturated fatty acids, MUFA = monounsaturated fatty acids, PUFA = polyunsaturated fatty acids.

**Table 6 foods-12-00966-t006:** The color measurement of the commercial fish protein concentrates, hydrolysates, and hydrolyzed collagen with mean and standard deviation of five replicate measurements. Different letters within the same row indicate a statistically significant difference with ANOVA post hoc pair comparison (*p* < 0.05). L* = lightness, a* = red-green, b* = yellow-blue.

Sample	L*	a*	b*	Whiteness
CONC1	68.4 ^b^ ± 0.44	3.3 ^b^ ± 0.06	19.9 ^b^ ± 0.19	62.5 ^b^ ± 0.30
CONC2	73.8 ^c^ ± 0.26	2.3 ^c^ ± 0.07	17.4 ^c^ ± 0.27	68.4 ^c^ ± 0.28
CONC3	84.6 ^a^ ± 0.15	0.1 ^a^ ± 0.03	13.3 ^a^ ± 0.05	79.6 ^a^ ± 0.12
HYDR1	80.2 ^d^ ± 0.10	2.7 ^d^ ± 0.26	27.0 ^d^ ± 0.23	66.4 ^d^ ± 0.17
HYDR2	87.5 ^e^ ± 0.21	1.7 ^e^ ± 0.04	21.1 ^e^ ± 0.11	75.4 ^e^ ± 0.14
COLL1	93.2 ^f^ ± 0.18	−1.8 ^f^ ± 0.04	13 ^a^ ± 0.14	85.2 ^f^ ± 0.21

Fish protein concentrates = CONC1–3, fish protein hydrolysates = HYDR1–2, hydrolyzed fish collagen = COLL1.

**Table 7 foods-12-00966-t007:** The sensory attributes of commercial fish protein concentrates, hydrolysates, and hydrolyzed collagen with mean and standard deviation of 27 (3 × 9) evaluations. Different letters within the same row indicate a statistically significant difference with ANOVA post hoc pair comparison (*p* < 0.05).

Attribute	HYDR1	HYDR2	COLL1	CONC1	CONC2	CONC3
Sedimentation	0.9 ± 1.1 ^a^	0.2 ± 0.4 ^a^	0.2 ± 0.4 ^a^	8.0 ± 0.8 ^b^	7.9 ± 0.9 ^b^	7.3 ± 1.0 ^b^
Color saturation	9.7 ± 0.6 ^a^	8.4 ± 1.0 ^a^	4.8 ± 0.4 ^b^	4.9 ± 1.8 ^b^	5.5 ± 1.9 ^b^	5.5 ± 1.2 ^b^
Cloudiness	1.2 ± 1.1 ^a^	0.5 ± 0.7 ^a^	0.4 ± 0.6 ^a^	9.3 ± 1.0 ^b^	9.2 ± 1.1 ^b^	8.6 ± 1.6 ^b^
Particle distinctiveness	0.1 ± 0.2 ^a^	0.1 ± 0.2 ^a^	0.1 ± 0.1 ^a^	4.8 ± 2.0 ^b^	6.0 ± 2.9 ^b^	4.7 ± 2.8 ^b^
Oily mouthfeel	1.8 ± 2.1 ^a^	0.5 ± 0.6 ^a^	0.4 ± 0.8 ^a^	1.9 ± 2.4 ^a^	1.8 ± 1.9 ^a^	1.7 ± 2.0 ^a^
Taste intensity	5.0 ± 2.3 ^a^	4.55 ± 2.4 ^a^	1.0 ± 0.9 ^b^	4.5 ± 2.3 ^a^	4.7 ± 2.2 ^a^	4.7 ± 2.4 ^a^
Sea/seaweed flavor	0.7 ± 0.6 ^a^	0.5 ± 0.6 ^a^	0.2 ± 0.3 ^a^	2.7 ± 1.9 ^b^	1.7 ± 1.1 ^c^	1.7 ± 1.3 ^c^
Metallic flavor	2.1 ± 2.7 ^ab^	3.1 ± 2.6 ^a^	1.1 ± 1.7 ^b^	1.2 ± 1.7 ^b^	1.0 ± 1.5 ^b^	0.8 ± 1.0 ^b^
Fishy flavor	3.7 ± 1.9 ^b^	2.0 ± 1.9 ^c^	0.4 ± 0.4 ^d^	4.7 ± 1.9 ^ab^	4.8 ± 2.1 ^ab^	5.3 ±1.8 ^a^
Bitterness	2.7 ± 2.3 ^a^	3.9 ± 2.0 ^a^	0.8 ± 1.2 ^b^	1.1 ± 1.4 ^b^	1.0 ± 1.4 ^b^	1.0 ± 1.4 ^b^
Umami flavor	3.3 ± 2.5 ^a^	2.7 ± 2.4 ^a^	0.4 ± 0.4 ^b^	1.1 ± 1.2 ^b^	1.2 ± 1.3 ^b^	1.2 ± 1.4 ^b^
Odor intensity	5.5 ± 2.1 ^a^	6.2 ± 1.9 ^a^	2.7 ± 2.4 ^b^	6.1 ± 1.6 ^a^	6.3 ± 1.5 ^a^	6.0 ± 1.8 ^a^
Fishy odor	4.1 ± 2.5 ^bc^	2.9 ± 2.9 ^cd^	1.3 ± 2.0 ^d^	6.7 ± 1.5 ^a^	6.7 ± 1.3 ^a^	6.2 ± 2.1 ^a^
Brothy odor	4.9 ± 2.6 ^b^	7.4 ± 1.9 ^a^	1.3 ± 1.5 ^c^	1.9 ± 2.0 ^c^	1.7 ± 2.3 ^c^	2.3 ± 2.3 ^c^
Sea/seaweed odor	1.1 ± 1.0 ^bc^	0.6 ± 0.9 ^c^	1.1 ± 1.1 ^bc^	2.0 ± 1.6 ^ab^	2.2 ± 1.4 ^a^	2.1 ± 1.6 ^a^

Fish protein concentrates = CONC1–3, fish protein hydrolysates = HYDR1–2, hydrolyzed fish collagen = COLL1.

**Table 8 foods-12-00966-t008:** Relative concentration (mean ± standard deviation) of different odor-active volatile compounds identified with GC-MS-O and previous literature from commercial fish protein concentrates, hydrolysates, and hydrolyzed collagen. The RI was determined with polar WF-Wax column. The identification is marked with RI+, if the compound could be identified also with non-polar DB-5 column. Different letters within the same row indicate a statistically significant difference with ANOVA post hoc pair comparison (*p* < 0.05).

#	Volatile Compound	RI	Identification	Odor Description	Relative Concentration (RC) ^1^ or Detection Frequency (DF) ^2^					
					CONC1	CONC2	CONC3	HYDR1	HYDR2	COLL1
1	Trimethylamine	<600	RI+, MS, O	Fishy, fatty	146 ± 16 ^a^	100 ± 57 ^a^	160 ± 79 ^a^	350 ± 75 ^b^	57 ± 15 ^a^	ND
2	Methanethiol	700	RI, MS, O	Cabbage, musty	ND ^3^	1 ± 1 ^a^	3 ± 1 ^a^	ND	5 ± 1 ^b^	3 ± 1 ^a^
3	Heptane	701	RI+, MS	- ^4^	9 ± 6 ^a^	14 ± 2 ^a^	2 ± 0 ^b^	ND	ND	ND
4	Propanal	801	RI, MS	-	ND	ND	27 ± 4 ^a^	3 ± 0 ^b^	ND	ND
5	3-Methylbutanal	929	RI+, MS, O	Sour bread	137 ± 9 ^a^	107 ± 27 ^a^	36 ± 8 ^b^	127 ± 26 ^a^	34 ± 4 ^b^	3 ± 1 ^b^
6	2-Ethylfuran	964	RI, MS	-	52 ± 2 ^ac^	50 ± 5 ^ac^	12 ± 1 ^a^	181 ± 44 ^b^	72 ± 9 ^c^	10 ± 1 ^a^
7	2,3-Butanedione	986	RI+, MS, O	Sweet, butter, toffee	13 ± 1 ^a^	8 ± 4 ^ac^	10 ± 2 ^a^	4 ± 1 ^bc^	1 ± 0 ^b^	ND
8	Unknown 1	1026		Glue, medicinal	NA ^5^	NA	DF 75 ^6^	DF 0	DF 0	NA
9	2,3-Pentanedione	1073	RI+, MS	Unclear	30 ± 7 ^a^	21 ± 4 ^ab^	15 ± 7 ^b^	ND	ND	ND
10	Dimethyl disulfide	1103	RI+, MS	-	ND	ND	ND	51 ± 12 ^a^	ND	71 ± 7 ^b^
11	Hexanal	1103	RI, MS, O	Grass	227 ± 101 ^a^	113 ± 33 ^bc^	39 ± 5 ^cd^	ND	16 ± 3 ^d^	ND
12	Unknown 2	1115		Coffee, solvent	NA	NA	DF 0	DF 75	DF 50	NA
13	1-Penten-3-ol	1170	RI+, MS	-	210 ± 10 ^a^	233 ± 37 ^a^	143 ± 26 ^b^	ND	ND	ND
14	2-Heptanone	1202	RI+, MS, O	Medicinal, burnt	31 ± 2 ^a^	39 ± 11 ^a^	4 ± 1 ^b^	21 ± 6 ^a^	3 ± 1 ^b^	1 ± 0 ^b^
15	Heptanal	1206	RI+, MS	-	121 ± 11 ^a^	94 ± 26 ^a^	11 ± 5 ^b^	14 ± 6 ^b^	6 ± 1 ^b^	2 ± 1 ^b^
16	2-Pentylfuran	1247	RI, MS	-	ND	6 ± 5 ^a^	ND	13 ± 5 ^ab^	20 ± 4 ^b^	5 ± 1 ^a^
17	(*Z*)-4-Heptenal	1263	RI, MS, O	Fishy, pungent	110 ± 8 ^a^	112 ± 15 ^a^	20 ± 4 ^b^	10 ± 3 ^bc^	1 ± 0 ^c^	ND
18	2-Octanone	1307	RI+, MS, O	Metallic	24 ± 4 ^a^	30 ± 4 ^a^	13 ± 4 ^b^	6 ± 2 ^bc^	1 ± 1 ^c^	ND
19	Octanal	1310	RI+, MS, O	Citrus, metallic	79 ± 11 ^a^	64 ± 23 ^a^	6 ± 4 ^b^	12 ± 3 ^b^	6 ± 1 ^b^	8 ± 1 ^b^
20	1-Octen-3-one	1324	RI, O	Mushroom	NA	NA	DF 100	DF 75	DF 0	NA
21	Hydroxyacetone	1337	RI, MS, O	Mushroom, meat broth	ND	ND	ND	ND	4 ± 1	NA
22	Unknown 3	1358		Dog food, leather	NA	NA	DF 0	DF 75	DF 0	NA
23	2-Acetyl-1-pyrroline	1375	RI+, O	Popcorn, Basmati rice	NA	NA	DF 100	DF 100	DF 50	NA
24	Unknown 4	1394		Floral, raspberry, green	NA	NA	DF 0	DF 0	DF 100	NA
25	Unknown 5	1396		Metallic, iron	NA	NA	DF 57	DF 0	DF 0	NA
26	Nonanal	1415	RI+, MS	-	65 ± 8 ^a^	58 ± 14 ^a^	12 ± 5 ^b^	41 ± 14 ^c^	44 ± 9 ^ac^	49 ± 4 ^ac^
27	2,3,5-Trimethylpyrazine	1435	RI+, MS, O	Musty	31 ± 3 ^a^	22 ± 11 ^a^	10 ± 2 ^b^	21 ± 7 *	1 ± 1 ^b^	ND
28	Dimethyl trisulfide	1436	RI+, MS	-	ND	ND	ND		ND	2 ± 0 ^a^
29	3-Isopropyl-2-metoxypyrazine	1438	RI, O	Soy sauce, green	NA	NA	22 ± 7 ^b^	ND	ND	ND
30	Unknown 6	1446		Ink, musty, burnt	NA	NA	DF 0	DF 0	DF 75	NA
31	1-Octen-3-ol	1449	RI, MS	-	77 ± 5 ^a^	89 ± 13 ^a^	22 ± 7 ^b^	ND	ND	ND
32	Acetic acid	1467	RI+, MS	-	ND	ND	ND	347 ± 150	ND	ND
33	Methional	1492	RI+, O	Boiled potato	NA	NA	DF 100	DF 75	DF 75	NA
34	2-Decanone	1514	RI+, MS	-	22 ± 2 ^ac^	34 ± 11 ^a^	2 ± 1 ^b^	11 ± 3 ^bc^	1 ± 0 ^b^	ND
35	2,4-Heptadienal	1531	RI, MS	-	4 ± 2 ^a^	2 ± 2 ^a^	ND	ND	ND	ND
36	Unknown 7	1543		Burnt, black currant	NA	NA	DF 0	DF 0	DF 100	NA
37	1-Octanol	1559	RI+, MS, O	Sour, pungent	17 ± 1 ^a^	19 ± 3 ^a^	5 ± 1 ^b^	ND	3 ± 1 ^b^	ND
38	Unknown 8	1595		Musty, green, pungent	NA	NA	DF 0	DF 75	DF 0	NA
39	2-Undecanone	1621	RI+, MS, O	Pungent	31 ± 4 ^a^	50 ± 13 ^b^	6 ± 2 ^c^	ND	ND	ND
40	Unknown 9	1649		Berry	NA	NA	DF 50	DF 50	NA	NA
41	2-Methylbutanoic acid	1687	RI, MS, O	Musty, sulfuric	ND	0.0	ND	33 ± 20 ^a^	6 ± 3 ^b^	ND
42	2-Dodecanone	1727	RI+, MS, O	Meat broth	ND	6 ± 6 ^a^	ND	1 ± 1 ^a^	ND	ND
43	Unknown 10	1757		Liquorice, cloying	NA	NA	DF 75	DF 75	DF 0	NA
44	Unknown 11	2074		Sweet, cotton candy	NA	NA	DF 50	DF 75	DF 100	NA
45	Unknown 12	2084		Rhubarb, acidic	NA	NA	DF 0	DF 0	DF 50	NA
46	Unknown 13	>2186		Solvent, ink	NA	NA	DF 50	DF 0	DF 0	NA

Fish protein concentrates = CONC1–3, fish protein hydrolysates = HYDR1–2, Hydrolyzed fish collagen = COLL1. * Same peak in chromatogram contained both 2,3,5-trimethylpyrazine and dimethyl trisulfide. Therefore, the relative concentration could not be determined, and the sample was left out from ANOVA for these two compounds. ^1^ Relative concentration was calculated as compound area equivalents compared to ISTD; ^2^ Detection frequency (n = 4) was reported if compound did not have a peak in the chromatogram or the compound was unknown; ^3^ ND = Not detected; ^4^ - = Not detected via GC-MS-O; ^5^ NA = Not analyzed via GC-MS-O; ^6^ DF = detection frequency (%).

## Data Availability

The data presented in this study are available on request from the corresponding author. The data are not publicly available due to the fact that the materials were provided by commercial operators.

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
