# Peer review of "Comparison of Commercial Fish Proteins’ Chemical and Sensory Properties for Human Consumption"

_foods, 2023, doi:10.3390/foods12050966_

Round 1

Reviewer 1 Report

The manuscript “Comparison of commercial fish proteins’ chemical and sensory properties for human consumption” is well planned and clearly written. A few comments and minor suggestions of changes are included.

Introduction

Line 55 – I suggest replacing “extracting” with “prepared”.

Line 65 – I think considering fish hydrolysates as fish proteins is not the most appropriate. It would be more adequate protein derivatives, for instance.

Materials and Methods

Lines 118 and 119 – I suggest including a quotation of the method followed in the determination of tryptophan content.

Line 181 – I think it is “Heated tubes were…”

Line 213 – I suggest “Thirty” instead of “30”.

Results and discussion

Lines 273-279 – Why ash content of analyzed products was not shown in Table 2? I also suggest including any comments to the ash content of these products. The protein content of COLL1 seems to be overestimated. Please note that the sum of the percentages of the four constituents of COLL1 accounts for 107.2 % and the ash content is not included.

Line 432 – The percentages of n-3 fatty acids presents in this line are not the same shown in Table 5. Please check.

Line 548-550 – This comment about the higher whiteness of protein hydrolysates compared to protein concentrates seems a bit exaggerated.

Line 553 – I think it is “Sathivel et al.66”. Please check.

Line 556 – Please check the end of this sentence because it is not clear.

Line 558 – It is “peroxide value” and not “peroxidase”, I suppose.

Figure 7 - It is “Peroxides” and not “Peroxidates”. Please check.

Reviewer 2 Report

This MS summarizes various commercial fish proteins' chemical and sensory properties (fish concentrates, hydrolysates, and fish collagen). The chemical and sensory properties of fish proteins have been evaluated from multiple perspectives, and the data quality is high. However, there are several points of concern.

Comments
(1) L32: The authors use the expression "sidestream", but do they mean by-product? In such a case, I would generally use "by-product". How about clarifying that you are using sidestram in the sense of by-product once you have clarified that you are using sidestream in the sense of by-product?
(2) L64-70: This study aims to characterize the chemical, functional and sensory properties of commercially available fish proteins. There are many more types of fish proteins on the market, so why were these five selected? I think it is necessary to indicate the reason in this section.
It would have been nice to analyze a wider variety of fish proteins.
(3) L89: I believe the nitrogen conversion factor for fish protein is 6.25, why is it 5.58?
(4) L261-263: Please describe the statistical processing method used to determine the normal distribution.
(5) Table 2: Table 2 does not show ash data.
(6) Table 5: There is an error in the abbreviation for MUFA. C16:1c-9 is correct for C16:1n-9. In addition, C16:1 is the main fatty acid of C16:1n-7 (palmitoleic acid). Is it really C16:1n-9?
(7) Fig 4-6: Please add a line to the vertical axis.
(8) Fig 6E: I think there is an error in the way you marked significant differences, please correct it. For example, the 20 min mark for the CONC1 group is “a”. Is that correct?
